# Cooperation of Experts: Fusing Heterogeneous Information with Large Margin

Shuo Wang [* 1]  Shunyang Huang [* 1]  Jinghui Yuan [* 2]  Zhixiang Shen [1]  Zhao Kang [† 1]

## Abstract

Fusing heterogeneous information remains a persistent challenge in modern data analysis. While significant progress has been made, existing approaches often fail to account for the inherent heterogeneity of object patterns across different semantic spaces. To address this limitation, we propose the **Cooperation of Experts (CoE)** framework, which encodes multi-typed information into unified heterogeneous multiplex networks. By transcending modality and connection differences, CoE provides a powerful and flexible model for capturing the intricate structures of real-world complex data. In our framework, dedicated encoders act as domain-specific experts, each specializing in learning distinct relational patterns in specific semantic spaces. To enhance robustness and extract complementary knowledge, these experts collaborate through a novel **large margin** mechanism supported by a tailored optimization strategy. Rigorous theoretical analyses guarantee the framework's feasibility and stability, while extensive experiments across diverse benchmarks demonstrate its superior performance and broad applicability. Our code is available at https://github.com/strangeAlan/CoE.

## 1. Introduction

Most real-world objects and data are heterogeneous in the form of multiple types or diverse forms of interactions, which pose significant challenges for modeling their intricate relationships (Han, 2009; Shen & Kang, 2025). For instance, multimodal data combines diverse sources, such as images and text, where each modality contributes unique features that collectively offer a richer understanding of the underlying phenomena; in a social network, people are linked with different types of ties: friendship, family relationship, professional relationship, etc. To characterize the complexities of this heterogeneity, we encode multi-typed information into a unified heterogeneous multiplex network, where each layer contains the same number of nodes (possibly with different attributes) but a different type of links, enabling a comprehensive and effective representation of the diverse relationships inherent in real-world data (Sun & Han, 2013).

Inspired by the success of graph neural networks (GNNs) (Kipf & Welling, 2016; Wang et al., 2019), numerous approaches have been proposed for multiplex network representation learning (Jing et al., 2021; Shen et al., 2024b). However, these methods often rely on training a single predictor across the entire network, which can lead to suboptimal results by neglecting the intrinsic heterogeneity and varying characteristics of different link patterns (Wang et al., 2024). As shown in Figures 1a and 1b, classifier performance, which is independently trained on each network, differs significantly across different layers, highlighting the importance of accounting for this variability. To effectively capture diverse node patterns, dedicated *experts*—components designed to focus on specific aspects of the network, are introduced (Shi et al., 2024). Each expert specializes in learning from a particular type of interaction, thereby reducing cross-interference between relationships and enabling a more nuanced representation of complex structures.

When utilizing multiple experts, effective collaboration among them becomes crucial. Existing Mixture of Experts (MoE) approaches employ a gating mechanism that activates only a subset of experts during both training and inference (Li et al.; Shi et al.). While this strategy improves efficiency, it inherently limits the ability to fully exploit the rich and diverse information embedded in heterogeneous data. This highlights the importance of fostering collaboration rather than competition among experts to leverage cross-layer knowledge effectively. However, this shift introduces two key challenges that must be addressed.

First, given the challenges of capturing the diverse and intricate patterns within networks, a key question arises: *how can we design a framework that effectively extracts and in-*

---

[*]Equal contribution [†]Corresponding author [1]University of Electronic Science and Technology of China, Chengdu, Sichuan Province, China [2]Northwestern Polytechnical University. Correspondence to: Shuo Wang <runner21st@gmail.com>, Zhao Kang <zkang@uestc.edu.cn>.

*Proceedings of the 42$^{nd}$ International Conference on Machine Learning*, Vancouver, Canada. PMLR 267, 2025. Copyright 2025 by the author(s).

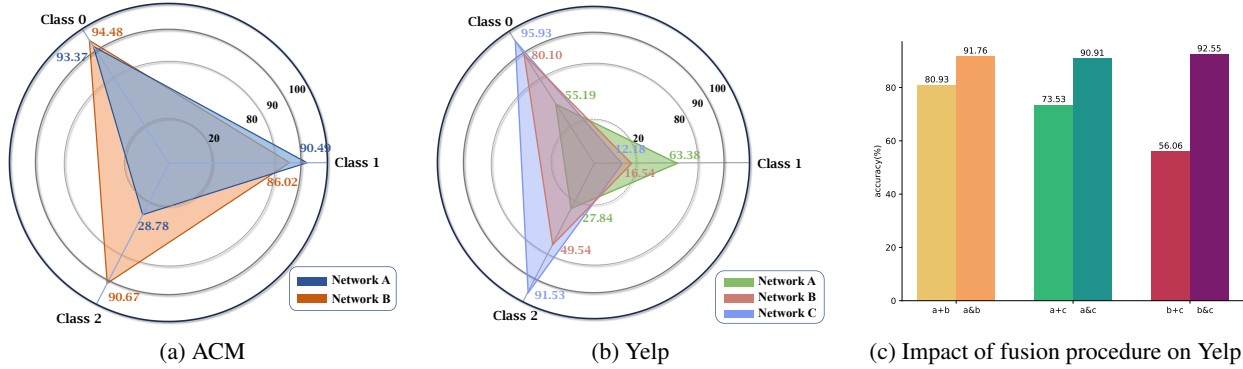

*Figure 1.* (a) and (b) present the classification results on different networks from the ACM and Yelp datasets, representing the diverse and intricate patterns within networks. (c) "+" symbol denotes directly adding the networks, while "&" represents the fusion procedure used in CoE.

*tegrates such complex information across all networks?* To address this, we propose **Cooperation of Experts (CoE)**, a novel architecture specifically designed to enable an efficient division of labor among experts. The framework adopts a two-level experts framework: low-level experts focus on learning patterns from individual networks, while high-level experts operate on a fused representation to capture cross-network relationships. Unlike simple concatenation, the fused representation is optimized by maximizing mutual information across networks. As shown in Figure 1c, this approach enhances performance by encoding complementary information, enabling more robust and comprehensive representations.

Second, *how can well-trained experts collaboratively contribute to the final prediction?* To promote cooperation rather than competition, we assign each expert a weighted influence in the decision-making process, determined by its proficiency in capturing distinct patterns. This is achieved through a learnable confidence tensor that dynamically adjusts the authority of each expert based on its specialization. To ensure effective collaboration, we introduce an innovative optimization mechanism for the confidence tensor, which strategically maximizes the **margin** between the two most confidently predicted outcomes. This mechanism not only balances the contributions of different experts but also addresses disparities in their expertise, ultimately boosting overall predictive performance.

Our primary contributions can be summarized as:

- We propose the **Cooperation of Experts (CoE)**, a novel expert coordination framework designed to extract stable and nuanced node patterns from multiplex networks. By tailoring experts to capture both specialized and shared features, CoE effectively disentangles and integrates heterogeneous information, uncovering intricate connections across diverse relationships.

- Our work further advances the utilization of multiple experts. To the best of our knowledge, this is the first framework to emphasize expert *cooperation* rather than competition. We introduce a novel **large margin** optimization strategy that maximizes the effectiveness of each expert's specialized expertise, fostering a more holistic and in-depth understanding of the underlying knowledge within networks.

- The effectiveness of our method is validated through rigorous theoretical analysis and extensive experimental evaluation. We conduct comprehensive experiments with various state-of-the-art methods on both multi-relational and multi-modal tasks.

## 2. Related Work

### 2.1. Multiplex Network Learning

Multiplex network learning focuses on capturing diverse structural patterns in multi-relational networks to support tasks such as node classification, clustering, and link prediction (Qian et al., 2024; Xie et al., 2025). With the rise of GNNs, HAN (Wang et al., 2019) integrates structural and attribute information using attention mechanisms at both the node and semantic levels, while MAGNN (Fu et al., 2020) and MHGCN (Yu et al., 2022) leverage graph convolution to extract relational features across multiple layers. MGBO (Huang et al., 2024) employs self-expression learning for homophily modeling, whereas HDMI (Jing et al., 2021) and DMG (Mo et al., 2023) utilize contrastive learning to enhance self-supervised representations. However, these methods heavily rely on reliable network topology structures, limiting their effectiveness in real-world scenarios with noisy or incomplete data.

## 2.2. Graph Structure Learning

Graph Structure Learning (GSL) has become a crucial area of research in GNNs, particularly for optimizing unreliable graph structures with significant noise. Supervised GSL methods often generate new adjacency matrices with learnable components, optimized using label information. These methods include probabilistic models (e.g., LDS and GEN (Franceschi et al., 2019; Wang et al., 2021)), similarity-based techniques (e.g., GRCN and IDGL (Yu et al., 2021; Chen et al., 2020)), attention mechanisms (e.g., NodeFormer (Wu et al., 2022)), or by treating all elements in the adjacency matrix as learnable parameters (e.g., ProGNN (Jin et al., 2020)). In contrast, methods like SUBLIME, STABLE, and GSR (Liu et al., 2022; Li et al., 2022; Zhao et al., 2023) leverage contrastive learning to construct graph structures in a self-supervised manner. While most previous studies have focused on single, homogeneous graphs, recent efforts have expanded GSL to heterogeneous graphs in supervised settings (Li et al., 2024a; Pan & Kang, 2023), as demonstrated by GTN and HGSL (Yun et al., 2019; Zhao et al., 2021). More recently, InfoMGF (Shen et al., 2024b) has explored unsupervised fusion of graphs from multiple sources, advancing research in multiplex graph structure learning. Despite these strides, there remains a lack of research addressing the challenge of effectively enabling the integration of complex information across multiple networks.

## 2.3. Multiple Learners Learning

Multiple Learners Learning is a powerful machine learning paradigm that leverages multiple models to enhance predictive performance. A classical approach within this paradigm is ensemble learning, which aggregates diverse models to reduce variance and bias, thereby improving generalization. Common ensemble methods include bagging (Zhou & Tan, 2024) and boosting (He et al., 2024). However, while boosting relies on sequential training where each model builds upon the previous one, it fails to independently capture information from diverse networks (Emami & Martínez-Muñoz, 2023). In contrast, advanced bagging techniques emphasize diversity among learners. Building on the classical Random Forest strategy (Rigatti, 2017), recent work introduces multiweighted bagging, such as the WRF method (Winham et al., 2013). Although ensemble learning facilitates information fusion, its advantages are predominantly observed in tabular data (Chen & Guestrin, 2016), making it challenging to extract complex patterns inherent in multiplex networks.

Another notable specialization within this domain is the expert mechanism. The classical Mixture of Experts (MoE) framework enhances adaptivity by assigning specialized submodels (i.e., experts) to distinct regions of the input space (Li et al.). A key limitation of many GNNs is their homogeneous processing of the entire graph, which overlooks structural and feature diversity. To address this, recent studies integrate MoE with graph learning (Wang et al., 2024; Kim et al., 2023) to improve adaptability. Furthermore, Mowst (Zeng et al., 2024) refines this approach by training experts to specialize in both feature and structure modalities. However, existing expert mechanisms primarily focus on expert competition, often neglecting the effective utilization of inter-relational dependencies—precisely the gap our work aims to bridge.

## 3. Preliminaries

**Heterogeneous Multiplex Networks.** It is represented by $G = \{G_1, ..., G_V\}$ composed of $V$ networks, where $G_v = \{A_v, X^v\}$ is the $v$-th network. $X^v \in \mathbb{R}^{N \times d_f}$ is the attribute matrix for $N$ nodes, so that $X_i \in \mathbb{R}^{d_f}$ represents the feature vector of node $i$. $A_v \in \{0, 1\}^{N \times N}$ is the corresponding adjacency matrix and $D_v$ is a diagonal matrix denoting the degree matrix of $A_v$. $Y$ denotes the node label. We summarize the notations in Appendix A.

**Experts Definition.** During the training phase, each expert is assigned a distinct task by focusing on specific layers, referred to as their *knowledge fields*. For instance, a classifier $F$ is exclusively trained on layer $G_1$ if $G_1$ constitutes the knowledge field of $F$. Consequently, experts are defined as the combination of classifiers and their corresponding layers within the designated knowledge fields, denoted as $E_i$.

## 4. Our Proposed Method

In this section, we introduce the CoE framework using a top-down approach. We begin by detailing the encoding of heterogeneous information into multiplex networks, followed by the two-level design of experts. Next, we describe the collaborative strategy employed by the experts, and finally, we conclude with the optimization of the critical confidence tensor. The overall framework of the proposed approach is shown in Figure 2.

### 4.1. Encoding Heterogeneous Information

Modeling arbitrary forms of heterogeneous information through multiplex networks presents a critical challenge: ensuring the reliability of the network structure. Real-world networks are often sparse and contain significant noise, which can degrade downstream performance. Therefore, we apply a Graph Structure Learning (GSL) strategy to refine and optimize the network structure.

Specifically, we employ a network learner, which not only generates refined networks but also ensures the preservation of crucial node features and structural information. We achieve this by utilizing the commonly employed Simple

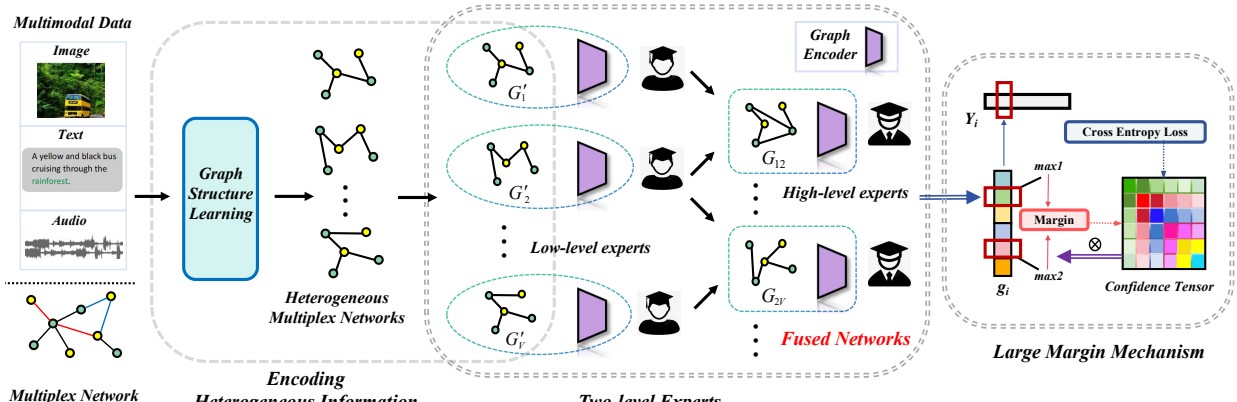

*Figure 2.* The overall framework of the proposed CoE. Specifically, CoE first encodes various information into heterogeneous multiplex networks, followed by network fusion through mutual information maximization. Subsequently, two-level experts are trained on single and fused networks respectively. Expert collaboration is enabled by a confidence tensor, which is optimized via a large margin mechanism.

Graph Convolution (SGC) (Wu et al., 2019), which is a two-layer attentive network:

$$H_v = \sigma(((\tilde{D}_v^{-\frac{1}{2}} \tilde{A}_v \tilde{D}_v^{-\frac{1}{2}})^r X^v)) \odot W_1^v) \odot W_2^v \quad (1)$$

where $\tilde{D}_v = D_v + I$ and $\tilde{A}_v = A_v + I$. $r$ represents the order of graph aggregation. $\sigma(\cdot)$ is the non-linear activation function and $\odot$ denotes the element-wise multiplication. For multi-modal data where $A_v$ is not provided, we use the original $X^v$ to replace $H_v$.

$H_v$ is subsequently utilized to reconstruct the network using the $K$-nearest neighbors ($KNN$) method. In practice, we employ $KNN$ sparsification with its locality-sensitive approximation to enhance efficiency (Fatemi et al., 2021). Once the reconstructed adjacency matrix is obtained, we perform post-processing operations, like existing GSL paradigm (Li et al., 2023), to ensure that the final adjacency matrix $A_v'$ adheres to key properties, including non-negativity, symmetry, and normalization. This process ultimately results in the construction of the heterogeneous multiplex networks $G = \{G_1', \cdots, G_V'\}$, where $G_i' = \{A_i', X^i\}$.

### 4.2. Two-level Design for Task-driven Experts

As described above, the two-level expert framework is designed to operate across distinct knowledge fields. Specifically, low-level experts focus on individual networks, learning network-specific link patterns, while high-level experts capture shared information across networks to identify higher-order dependencies that cannot be uncovered when networks are analyzed independently. To enhance the task relevance of low-level experts, we guide their learning by formulating it as the maximization of $I(G_i'; Y)$. For high-level experts, we incorporate $I(G_i'; G_j')$ as a maximization objective during the training phase, which has been theoretically shown to provide a lower bound for

$I(G_i'; G_j) + I(G_i; G_j')$ (Federici et al.). The loss function can be formulated as:

$$\mathcal{L}_E = -\sum_{i=1}^{V} I(G_i'; Y) - \sum_{i=1}^{V} \sum_{j=i+1}^{V} I(G_i'; G_j')$$
$$- \sum_{i=1}^{V} I(G_i'; G_{tot}) - \sum_{i=1}^{V} \sum_{j \neq i} I(G_i'; G_{ij}) \quad (2)$$

where we fuse $G_i'$ and $G_j'$ by optimizing the loss function above to generate the fused network $G_{ij}$. Additionally, an extra high-level expert is trained on $G_{tot}$, which is obtained by fusing all refined single networks following the same fusion procedure. Through mutual information maximization, $G_{ij}$ effectively captures the intricate information shared across the single networks. Since directly computing mutual information is impractical due to the complexity of network-structured data, we reformulate the network-level mutual information in terms of node-level representations.

**Theorem 4.1.** *Given a network $G$ with label $Y$, the cross-entropy loss $\mathcal{L}_{cls}(Z, Y)$ is the upper bound of $-I(G; Y)$, where $Z$ denotes the node representations of all nodes in network $G$. (Sun et al., 2022; Li et al., 2024b)*

Following Theorem 4.1, the original $\mathcal{L}_E$ could be eventually transformed as follows:

$$\hat{\mathcal{L}}_E = \sum_{i=1}^{V} \mathcal{L}_{cls}(Z^i; Y) - \sum_{i=1}^{V} \sum_{j=i+1}^{V} I_{lb}(Z^i; Z^j)$$
$$- \sum_{i=1}^{V} I_{lb}(Z^i; Z^{tot}) - \sum_{i=1}^{V} \sum_{j \neq i} I_{lb}(Z^i; Z^{ij}) \quad (3)$$

Here, $I_{lb}(Z^i; Z^j)$ is the lower bound of the mutual information $I(Z^i; Z^j)$ between networks $i$ and $j$ (Liang et al.,

2024):

$$I_{lb}(Z^i; Z^j) = \mathbb{E}_{\substack{z^i, z^{j+} \sim p(z^i, z^j) \\ z^j \sim p(z^j)}} \left[ log \frac{exp f(z^i, z^{j+})}{\sum_N exp f(z^i, z^j)} \right]$$
(4)

where $f(\cdot, \cdot)$ is a score critic approximated using a non-linear projection and cosine similarity. The joint distribution of node representations from networks $i$ and $j$ is denoted by $p(z^i, z^j)$, while the marginal distribution is represented by $p(z^i)$. The representations $z^i$ and $z^{j+}$ are positive samples of each other, indicating that they correspond to the same node in networks $i$ and $j$, respectively. Implementation details are summarized in Appendix E.

### 4.3. Large Margin Mechanism

Since each expert demonstrates a distinct knowledge preference for different node patterns, it is crucial to bridge the information gap across various knowledge fields. Unlike traditional expert coordination mechanisms (Chen et al., 2022), which rely on a subset of experts, we advocate leveraging all experts in the decision-making process. This approach reduces the risk of amplifying noise caused by the reliance on a limited number of experts. To promote collaboration rather than competition among experts, we introduce a confidence tensor, defined as follows.

**Definition 4.2.** A confidence tensor $\Theta \in \mathbb{R}^{c \times c \times k}$ is defined, where $c$ denotes the number of categories and $k$ represents the number of experts. The $(r, s, t)$-th element of $\Theta$, denoted as $\Theta_{rst}$, quantifies the credibility assigned by the $t$-th expert to the prediction that a sample belongs to the $r$-th category when it actually belongs to the $s$-th category.

For the convenience of calculation, we empirically expand the confidence tensor $\Theta$ into $\mathbb{R}^{c \times kc}$. Therefore, the final collaborative prediction result of the experts is obtained according to the following formula

$$\hat{y}_i = \underset{j=1...c}{argmax} \ (\mathcal{S}(\Theta g_i))_j$$
(5)

where $g_i$ is the collection of various expert opinions. For any expert $E_j$ and a given node $v_i$, the corresponding decision could be represented as a vector $E_j(v_i) \in \mathbb{R}^{c \times 1}$, thus $g_i$ is $(E_1^\top(v_i), E_2^\top(v_i), \cdots, E_k^\top(v_i))^\top$. $\mathcal{S}(\cdot)$ represents the softmax operation on the internal vectors.

By leveraging the confidence tensor, all experts contribute to the final decision-making process with varying levels of influence. However, in a multi-expert collaborative system, significant errors within the cognitive framework of a particular expert have the potential to mislead the entire system. To mitigate this risk and improve generalization, it is crucial to design a loss function to encourage consistency among experts. This loss function is designed to maximize the margin between the most confidently predicted outcome

and the second most confidently predicted outcome among experts. By doing so, it ensures greater consistency in predictions made by different experts, enhancing the overall robustness and reliability of the system. This objective can be formalized as follows:

$$Y_i^T(Y_i \odot \mathcal{S}(\Theta g_i)) - max_2(\mathcal{S}(\Theta g_i))$$
(6)

with the symbol $max_2(v)$ representing the second largest element in vector $v$, $Y_i$ is the $i$-th column of $Y$, while the symbol $\odot$ represents the Hadamard product. It is worth noting that the definition of margin itself is given by $max_1(\mathcal{S}(\Theta g_i)) - max_2(\mathcal{S}(\Theta g_i))$, where $max_1(v)$ is the highest value in vector $v$. It is proved that using our defined margin (Eq. 6) leads to the same optimization direction as the original definition while avoiding approximations (Yuan et al., 2024).

However, the max function is non-smooth, and the $max_2$ function is both non-convex and non-smooth. Therefore, we use the logsumexp function to approximate the max function, which is expressed as $f(v) = \frac{1}{\alpha} log \left( \sum_{j=1}^c e^{\alpha v_j} \right)$, where $\alpha$ is a quite large constant. By setting the maximum value of the vector to be operated upon to zero and then applying the max operator, $max_2$ function can be effectively computed by:

$$max_2(\mathcal{S}(\Theta g_i)) = \frac{1}{\alpha} log \left( \sum_{j=1}^c e^{\alpha(\mathcal{S}(\Theta g_i) - Y_i \odot \mathcal{S}(\Theta g_i))_j} \right)$$
(7)

By adding up the loss term for each node, the generalization loss is:

$$\mathcal{M} = \sum_{i=1}^N Y_i^\top(Y_i \odot \mathcal{S}(\Theta g_i))$$
$$- \sum_{i=1}^N \frac{1}{\alpha} log \left( \sum_{j=1}^c e^{\alpha(\mathcal{S}(\Theta g_i) - Y_i \odot \mathcal{S}(\Theta g_i))_j} \right)$$
(8)

On the other hand, it is also crucial to ensure that predictions from all experts always tend to be correct. We use cross-entropy loss to measure the distance from the correct opinion.

$$\mathcal{C} = - \sum_{i=1}^N (Y_i^\top(Y_i \odot log(\mathcal{S}(\Theta g_i))))$$
(9)

Thus two types of losses are incorporated during the tensor optimization phase, where $\mathcal{C}$ and $\mathcal{M}$ are specifically designed to enhance the correctness and consistency of expert decisions. Accordingly, the overall loss function can be

formulated as

$$\mathcal{L} = \mathcal{C} - \eta\mathcal{M} + \sum_{i=1}^{V} \mathcal{L}_{cls}(Z^i, Y) - \sum_{i=1}^{V}\sum_{j=i+1}^{V} I_{lb}(Z^i; Z^j)$$

$$- \sum_{i=1}^{V} I_{lb}(Z^i; Z^{tot}) - \sum_{i=1}^{V}\sum_{j\neq i} I_{lb}(Z^i; Z^{ij})$$

$$(10)$$

where $\eta$ is a given hyperparameter to balance $\mathcal{C}$ and $\mathcal{M}$, additional experiments are conducted to show the impact of all parameters appeared in the optimization phase.

## 5. Theoretical Analysis

In this section, we establish that our proposed model satisfies key properties, including partial convexity, Lipschitz continuity and robust generalization ability. Additionally, we present the optimization algorithm and prove its convergence to critical points of the model's loss function. Partial convexity indicates that the loss function remains convex with respect to certain intermediate variables, ensuring a smoother optimization landscape and reducing the risk of getting trapped in local optima.

**Theorem 5.1.** $\mathcal{L}(\Theta g_i)$ *is a convex function with respect to* $(\Theta g_i)$.

**Theorem 5.2.** $\mathcal{L}$ *is Lipschitz continuous, and the Lipschitz constant* $L \leq 2\sqrt{c}k\left(1 + \gamma + \frac{\gamma}{c}e^{\alpha}\right)$.

Consider performing gradient descent for $T$ iterations, yielding a sequence of points $\{\Theta_0, \Theta_1, \ldots, \Theta_t, \ldots, \Theta_{T-1}\}$. Due to the Lipschitz continuity of the loss function, the norm of the gradient at the point with the smallest gradient converges to zero. This implies that, with a sufficiently large number of iterations, there must be at least one critical point within the set of points visited during the process.

**Theorem 5.3.** *For a step size* $\eta \leq \frac{1}{L}$, *the gradient descent algorithm generates a sequence* $\{\Theta_t\}$ *such that*

$$\min_{t=0,1,\ldots,T} \|\nabla\mathcal{L}(\Theta_t)\|^2 \leq \frac{2(\mathcal{L}(\Theta_0) - \mathcal{L}^*)}{\eta T} \qquad (11)$$

*where* $\mathcal{L}^*$ *is the minimum value of* $\mathcal{L}(\Theta)$.

The above theorem proves that CoE is theoretically convergent. All proofs for theorems in this section are given in Appendix D.

Moreover, as generalization capability serves as a critical metric for evaluating model performance, we conduct a systematic analysis of the proposed model's generalization behavior in this study.

We begin by formalizing the learning setup. Let $\mathcal{X} \times \mathcal{Y}$ be the input-label space with $|\mathcal{Y}| = C$ classes. We have

an i.i.d. training sample $S = \{(x_i, y_i)\}_{i=1}^n$. A CoE classifier $f : \mathcal{X} \to \Delta^C$ produces a probability vector over $C$ classes, denoted $f(x) = (f_1(x), \ldots, f_C(x))^\top$. We define the margin of $f$ at $(x, y)$ as

$$\gamma_f(x, y) := f_y(x) - \max_{y'\neq y} f_{y'}(x), \qquad (12)$$

a large positive $\gamma_f(x, y)$ implies a strong preference for the correct class $y$. The usual 0-1 loss is $\ell_{0\text{-}1}(f; x, y) := \mathbb{I}[\arg\max_c f_c(x) \neq y]$. We also define $\ell_\gamma^{0\text{-}1}(f; x, y) := \mathbb{I}[\gamma_f(x, y) \leq 0]$ and the ramp loss:

$$\ell_\gamma(f; x, y) := \begin{cases} 0, & \gamma_f(x, y) \geq \gamma, \\ 1 - \dfrac{\gamma_f(x, y)}{\gamma}, & 0 < \gamma_f(x, y) < \gamma, \quad (13) \\ 1, & \gamma_f(x, y) \leq 0. \end{cases}$$

One has $\ell_{0\text{-}1}(f; x, y) \leq \ell_\gamma^{0\text{-}1}(f; x, y) \leq \ell_\gamma(f; x, y)$.

Let $\ell_\gamma \circ \mathcal{F}$ be the set of ramp-loss functions induced by a hypothesis class $\mathcal{F}$, where each $f \in \mathcal{F}$ is a CoE classifier. Then for any $\delta > 0$, with probability at least $1 - \delta$ over an i.i.d. sample $S = \{(x_i, y_i)\}_{i=1}^n$, the following holds for all $f \in \mathcal{F}$:

$$\mathbb{E}[\ell_{0-1}(f)] \leq \mathbb{E}_{(x,y)\sim\mathcal{D}}[\ell_\gamma^{0-1}(f; x, y)]$$

$$\leq \frac{1}{n}\sum_{i=1}^{n} \ell_\gamma(f; x_i, y_i) + \frac{2}{\gamma}\mathfrak{R}_n(\mathcal{F}) + 3\sqrt{\frac{log(\frac{2}{\delta})}{2n}}, \quad (14)$$

where $\mathfrak{R}_n(\mathcal{F})$ is the Rademacher complexity of the CoE margin-function class.

In CoE framework, we have $k$ experts $E_1, \ldots, E_k$, each outputs a probability vector $E_j(x) \in \mathbb{R}^C$. Besides, we have a confidence tensor $\Theta$ and we form $g(x) = [E_1(x)^\top, \ldots, E_k(x)^\top]^\top$, then $f(x) = \text{softmax}(\Theta g(x))$. And the margin is $\gamma_f(x, y) = [\Theta g(x)]_y - \max_{y'\neq y}[\Theta g(x)]_{y'}$.

Here we assume $\|\Theta\|_F \leq B_\Theta$ and $\|E_j(x)\|_2 \leq G_e, \forall j, x$, where $B_\Theta$ and $G_e$ could be considered as two bounds. Hence $\|g(x)\|_2 \leq \sqrt{k}\,G_e$. Let $\mathcal{F}_\Theta$ be the set of CoE margin functions $\gamma_f$.

Then $\mathfrak{R}_n(\mathcal{F}_\Theta) \leq C_{\text{MC}}\frac{B_\Theta G_e\sqrt{k}}{\sqrt{n}}$, where $C_{\text{MC}}$ is a constant on the order of $\sqrt{ln(C)}$, reflecting the multi-class max operation. With probability at least $1 - \delta$, all $f \in \mathcal{F}_\Theta$ satisfy

$$\mathbb{E}\big[\ell_{0\text{-}1}(f)\big] \leq \frac{1}{n}\sum_{i=1}^{n} \ell_\gamma(f; x_i, y_i) + \frac{2B_\Theta G_e\sqrt{k}}{\gamma\sqrt{n}} + 3\sqrt{\frac{log(\frac{2}{\delta})}{2n}}.$$

$$(15)$$

*Table 1.* Accuracy ± STD comparison (%) under the setting of network node classification task. The highest result is highlighted with **red boldface** and the runner-up is **bolded**. The symbol "OOM" means out of memory.

| Methods | ACM | DBLP | Yelp | MAG | Amazon |
|---|---|---|---|---|---|
| GCN | 89.04±0.62 | 80.70±0.30 | 74.03±1.61 | 74.60±0.13 | 93.12±0.28 |
| HAN | 91.30±0.33 | 81.28±0.23 | 52.04±1.55 | OOM | OOM |
| LDS | 88.55±0.42 | 87.17±0.81 | 87.11±1.99 | OOM | OOM |
| GRCN | 92.00±0.20 | 91.10±0.43 | 78.30±0.99 | OOM | 95.27±0.31 |
| IDGL | 92.33±1.17 | 79.29±0.58 | 88.28±5.08 | OOM | 93.68±0.45 |
| ProGNN | 92.09±1.21 | 91.10±1.46 | 60.43±1.36 | OOM | 93.12±0.47 |
| GEN | 90.82±2.69 | 91.33±0.70 | 69.22±3.04 | OOM | 97.51±1.05 |
| NodeFormer | 90.73±0.88 | 80.26±0.66 | 90.76±1.03 | 77.14±0.19 | 97.72±0.38 |
| SUBLIME | 91.81±0.26 | **91.49±0.32** | 90.91±0.75 | 67.45±0.22 | 97.02±0.93 |
| STABLE | 89.18±4.41 | 79.00±1.98 | 89.77±3.36 | OOM | 97.76±0.51 |
| GSR | 91.39±1.17 | 77.83±0.38 | 84.46±0.79 | OOM | 94.10±0.79 |
| HDMI | 91.45±0.43 | 90.14±0.38 | 73.75±0.87 | 69.25±0.12 | 95.08±0.47 |
| InfoMGF | **92.81±0.29** | 91.45±0.37 | **92.01±0.42** | 77.32±0.09 | 97.78±0.21 |
| GMoE | 90.29±0.41 | 91.18±0.43 | 91.92±0.37 | 77.27±0.54 | 97.78±0.41 |
| Mowst | 85.69±1.01 | 89.69±0.99 | 91.31±1.17 | **77.40±0.11** | **97.89±0.44** |
| CoE | **94.21±0.14** | **92.27±0.24** | **93.40±0.12** | **78.37±0.16** | **98.01±0.09** |

It shows that ensuring a large margin $\gamma$ and controlling the norms $B_\Theta$ (confidence-tensor magnitude) and $G_e$ (expert-output scale) leads to a generalization guarantee. Increasing the number of experts $k$ has a $\sqrt{k}$ impact, illustrating the trade-off between model capacity and margin-based guarantees. Due to CoE mechanism limits the number of $k$ and $B_\Theta$, while $G_e$ is fixed, thus our model has remarkable generalization ability.

# 6. Experiments

In this section, we present comprehensive experiments to thoroughly evaluate the effectiveness and robustness of our proposed CoE model. Specifically, we address the following research questions: **RQ1:** Does CoE outperform SOTA models on multiplex and multimodal network datasets? **RQ2:** What is the impact of the key components on model performance? **RQ3:** How robust is the CoE algorithm when subjected to structural attacks or noise?

## 6.1. Experimental Setup

**Datasets.** To thoroughly evaluate the effectiveness of CoE, we conduct experiments on five benchmark network datasets, including citation networks (ACM (Yun et al., 2019) and DBLP (Yun et al., 2019)), review networks (Yelp (Lu et al., 2019) and Amazon (McAuley & Leskovec, 2013; Gao et al., 2023)), and a large-scale citation network MAG (Wang et al., 2020). Additionally, we perform experiments on four multimodal datasets: ESP, Flickr, IAPR, and NUS (Xia et al., 2023), which lack structural information. Thus, we first build network structures with $K$NN for them. De-

tailed descriptions of these datasets can be found in Appendix C.

**Baselines.** We evaluate the performance of our method by comparing it against a range of existing approaches. In the network data scenario, we select two supervised structure-fixed GNNs—GCN (Kipf & Welling, 2016) and HAN (Wang et al., 2019)—as well as six supervised GSL methods: LDS (Franceschi et al., 2019), GRCN (Yu et al., 2021), IDGL (Chen et al., 2020), ProGNN (Jin et al., 2020), GEN (Wang et al., 2021), and NodeFormer (Wu et al., 2022). Additionally, we include three unsupervised GSL methods: SUBLIME (Liu et al., 2022), STABLE (Li et al., 2022), and GSR (Zhao et al., 2023), and two unsupervised multiplex methods: HDMI (Jing et al., 2021) and InfoMGF (Shen et al., 2024b). Finally, we also compare against two state-of-the-art Graph-MoE methods: GMoE (Wang et al., 2024) and Mowst (Zeng et al., 2024).

For multimodal data, we select four representative GSL methods, including two supervised approaches (ProGNN and GEN) and two unsupervised approaches (SUBLIME and InfoMGF). Additionally, we incorporate three multiview methods (i.e., DCCAE (Wang et al., 2015), CPM-Nets (Zhang et al., 2020), and ECML (Xu et al., 2024)) and two multimodal methods (i.e., MMDynamics (Han et al., 2022) and QMF (Zhang et al., 2023)). Classification accuracy is used as the evaluation metric. To ensure a fair comparison, we adopt GCN as the backbone encoder for all models. For methods that cannot support multiplex networks or multimodal data learning, we conduct training on each network or modality separately and report the maximum accuracy results. Implementation details can be found in Appendix

B.

## 6.2. Node Classification Performance (RQ1)

The experimental results, presented in Table 1, show that our proposed CoE achieves the highest accuracy across all datasets, consistently outperforming existing approaches. Notably, CoE surpasses recent Graph-MoE and multiplex methods. Furthermore, our method exhibits the lowest standard deviation in most cases, highlighting its stability.

The multimodal data classification results are summarized in Table 2, where the topology is first inferred from the initial feature matrix. As shown in the table, our proposed CoE significantly outperforms baseline models, including both network learning frameworks and multimodal approaches. These results highlight CoE's stability and adaptability across diverse settings.

*Table 2.* Accuracy $\pm$ STD comparison ($\% \pm \sigma$) under the setting of multimodal data classification task.

| Methods | ESP | Flickr | IAPR | NUS |
|---------|-----|--------|------|-----|
| ProGNN | 78.52±0.16 | 67.77±0.24 | 67.63±0.11 | 62.78±0.39 |
| GEN | 79.05±0.24 | 63.33±0.22 | 67.75±0.29 | 64.26±0.19 |
| SUBLIME | 77.53±0.14 | 64.97±0.18 | 62.06±0.11 | 52.54±0.16 |
| InfoMGF | 79.23±0.24 | 68.79±0.22 | 68.75±0.18 | 63.80±0.19 |
| DCCAE | 78.48±0.45 | 67.98±0.24 | 60.73±0.64 | 62.39±0.27 |
| CPM-Nets | 80.09±0.59 | **69.49±0.46** | 67.33±0.45 | 65.34±0.54 |
| ECML | 71.21±0.11 | 62.41±0.16 | 58.89±0.18 | 64.34±0.14 |
| MMDynamics | **80.19±0.56** | 65.44±0.73 | 66.59±0.41 | 63.64±0.68 |
| QMF | 80.14±0.34 | 69.24±0.34 | **69.08±0.45** | **65.42±0.37** |
| CoE | **81.11±0.05** | **70.24±0.09** | **71.04±0.04** | **66.80±0.32** |

## 6.3. Ablation Studies (RQ2)

To assess the contribution of each component in the CoE framework, we design four variant models by systematically modifying or removing specific elements and evaluate their performance on the node classification task. The experimental results, presented in Table 3, provide insights into the impact of each component on overall performance.

*Effectiveness of the expert learning components.* Recall that CoE employs GSL with a two-level expert design, incorporating both low-level and high-level experts. To analyze their contributions, we construct two variant models: one without high-level experts (w/o-HE) and another without the GSL process (w/o-GSL). The results show that both variants exhibit degraded performance, with the absence of high-level experts having a more pronounced negative impact compared to the removal of GSL.

*Effectiveness of expert collaboration mechanism.* To evaluate the effectiveness of the proposed large-margin mecha-

*Table 3.* Performance ($\% \pm \sigma$) of CoE and its variants.

| Variants | ACM | DBLP | Yelp | MAG | Amazon |
|----------|-----|------|------|-----|--------|
| RF | 93.39±0.19 | 91.48±0.14 | 91.61±0.21 | 76.91±0.16 | 97.56±0.12 |
| WRF | 93.64±0.15 | 91.97±0.22 | 93.05±0.16 | 77.45±0.18 | 97.78±0.26 |
| w/o HE | 91.25±0.17 | 90.71±0.26 | 68.27±0.32 | 78.05±0.27 | 94.50±0.16 |
| w/o GSL | 93.60±0.29 | 91.13±0.24 | 93.14±0.33 | 77.97±0.34 | 97.87±0.24 |
| CoE | 94.21±0.14 | 92.27±0.24 | 93.40±0.12 | 78.37±0.16 | 98.01±0.09 |

nism, we replace it with two variants: the original random forest (RF) (Rigatti, 2017) and the weighted random forest (WRF) (Winham et al., 2013). Experimental results confirm the superiority of our large-margin mechanism. Notably, WRF outperforms RF, suggesting that different experts demonstrate varying levels of proficiency for different node patterns. This finding highlights the importance of designing a tailored expert collaboration mechanism to accommodate specific scenarios.

## 6.4. Robustness Analysis (RQ3)

To assess the robustness of our proposed method, we introduce perturbations to the network structure by randomly adding or removing edges in the ACM dataset. The proportion of modified edges is varied from 0 to 0.9 to simulate different levels of attack intensity. For comparison, we evaluate several baseline methods, including the traditional fixed-structure model (GCN), the graph structure learning method (SUBLIME), the multiplex learning method (InfoMGF), and the expert-mixing approach (Mowst).

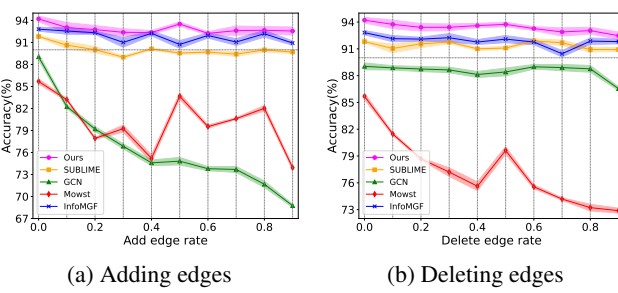

(a) Adding edges      (b) Deleting edges

*Figure 3.* Robustness analysis on ACM.

The experimental results, presented in Figure 3, clearly show that as perturbations increase, the performance of all methods degrades to varying extents. However, graph structure learning methods (CoE, InfoMGF, and SUBLIME) exhibit strong resilience. In particular, CoE benefits significantly from GSL, allowing it to accurately detect and correct disrupted network connections. Even under extreme perturbations, CoE consistently outperforms other methods, demonstrating exceptional robustness in maintaining stable performance and effectively mitigating the adverse effects of structural distortions.

Furthermore, we investigate the impact of two key hyperparameters in CoE: the optimization-phase parameters $\gamma$ and $\alpha$. We set these parameters to $\{50, 100, 200, 500, 1000\}$. As illustrated in Figure 4, CoE shows low sensitivity to variations in both $\gamma$ and $\alpha$, demonstrating the remarkable stability of our model. Additional sensitivity analyses for other hyperparameters are provided in Appendix F.

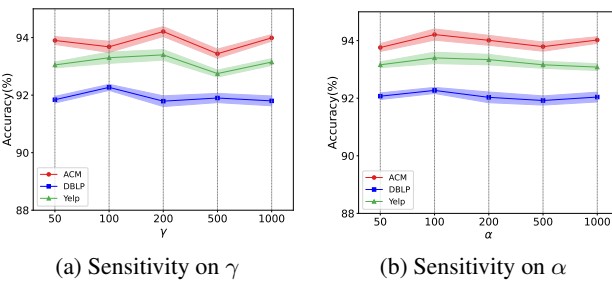

(a) Sensitivity on $\gamma$      (b) Sensitivity on $\alpha$

*Figure 4.* Sensitivity analysis on critical hyper-parameters.

## 7. Conclusion

In this work, we propose and analyze a novel paradigm for fusing diverse information through heterogeneous multiplex networks. Our approach introduces an expert mechanism into network learning and provides a tailored optimization strategy. We present a comprehensive theoretical analysis alongside extensive experiments, comparing our method with various baselines. To the best of our knowledge, this is the first study to explore expert learning in multiplex networks. Unlike traditional approaches that emphasize expert competition, our work pioneers a shift toward expert cooperation, advocating for further research on applying the Collaboration of Experts mechanism to more complex scenarios.

## Impact Statement

This paper aims to advance the field of Machine Learning. While our work has potential societal implications, we do not identify any specific concerns that require particular emphasis at this stage.

## Acknowledgments

This work was supported by the National Natural Science Foundation of China (No. U24A20323).

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

# A. Notations

*Table 4.* Notations used in our paper.

| Notation | Description |
|---|---|
| $V, N, d_f$ | The number of networks/nodes/features. |
| $G = \{G_1, ..., G_V\}$ | The heterogeneous multiplex network. |
| $G_v = \{A_v, X^v\}$ | The $v$-th original network. |
| $A_v \in \{0,1\}^{N \times N}$ | The adjacency matrix of $v$-th original network. |
| $D_v$ | The diagonal matrix denoting the degree matrix of $A_v$. |
| $X^v \in \mathbb{R}^{N \times d_f}$ | The attribute matrix of $v$-th network. |
| $X_i \in \mathbb{R}^{d_f}$ | The feature vector of node $i$. |
| $G'_v = \{A'_v, X^v\}$ | The $v$-th refined network. |
| $G_{tot}$ | The network fused by all refined single networks. |
| $E_i$ | The expert integrates the node predictor and the corresponding networks in its knowledge field. |
| $Y$ | The label information. |
| $H^v \in \mathbb{R}^{N \times d_f}$ | The node embeddings of the original network from the attention learner. |
| $r$ | The order of graph aggregation. |
| $K$ | The number of neighbors in $K$NN. |
| $c$ | The number of categories. |
| $g_i$ | The collection of various expert opinions. |
| $k$ | The number of experts. |
| $\Theta$ | The confidence tensor. |
| $I(G'_i; G'_j)$ | The mutual information between the $i$-th and $j$-th refined networks. |
| $\mathcal{M}$ | The generalization loss contributed by all nodes. |
| $\mathcal{C}$ | The cross-entropy loss to measure the distance from the correct opinion. |
| $\mathcal{L}_{cls}$ | The cross-entropy loss. |
| $\mathcal{L}_E$ | The training loss of all experts. |
| $\mathcal{L}$ | The overall loss function. |
| $max_1(v)$ | The largest element in vector $v$. |
| $max_2(v)$ | The second largest element in vector $v$. |
| $\alpha$ | Constant used in computing $max_2$. |
| $\eta$ | A given hyperparameter in $\mathcal{L}$. |
| $f(\cdot, \cdot)$ | The score critic approximated using a non-linear projection and cosine similarity. |
| $\mathcal{S}(\cdot)$ | The softmax operation on the internal vectors. |
| $\sigma(\cdot)$ | The non-linear activation function. |
| $\odot$ | The Hadamard product. |

# B. Hyper-parameters Settings and Infrastructure

*Table 5.* Details of the hyper-parameters settings.

| Dataset | $E$ | $lr$ | $d_h$ | $d$ | $K$ | $r$ | $L$ | $\tau_c$ | $\alpha$ | $\gamma$ |
|---|---|---|---|---|---|---|---|---|---|---|
| ACM | 800 | 0.001 | 128 | 64 | 15 | 2 | 2 | 0.2 | 100 | 100 |
| DBLP | 400 | 0.005 | 64 | 32 | 10 | 2 | 2 | 0.2 | 100 | 100 |
| Yelp | 400 | 0.0001 | 128 | 64 | 15 | 2 | 2 | 0.2 | 100 | 100 |
| MAG | 400 | 0.001 | 256 | 64 | 15 | 3 | 3 | 0.2 | 100 | 100 |
| Amazon | 400 | 0.001 | 256 | 64 | 15 | 3 | 3 | 0.2 | 100 | 100 |

All experiments are conducted on a platform equipped with an Intel(R) Xeon(R) Gold 5220 CPU and an NVIDIA A800 80GB GPU, using PyTorch 2.1.1 and DGL 2.4.0. Each experiment is run five times and the average results are reported.

Our model is trained using the Adam optimizer. The hyper-parameter settings for all datasets are presented in Table 5. Here, $E$ represents the number of training epochs, which is tuned within the set $\{100, 200, 400, 500, 800, 1000\}$ and $lr$ is the learning rate which is searched within the set $\{0.0001, 0.005, 0.001, 0.005, 0.01\}$. The hidden layer dimension $d_h$ and the representation dimension $d$ of the graph encoder GCN are tuned within the set $\{32, 64, 128, 256\}$. The number of neighbors $K$ for $K$NN is searched within the set $\{5, 10, 15, 20, 30\}$. The order of graph aggregation $r$ and the number of layers $L$ in GCN are set to either 2 or 3, which is in line with the common layer count of GNN models (Baranwal et al.). The temperature parameter $\tau_c$ in the contrastive loss is fixed at 0.2. In the optimization phase, the parameters $\alpha$ and $\gamma$ are

both set to 100. For the large-scale datasets MAG and Amazon, when computing the contrastive loss for estimating mutual information, we use a batch size of 2560 and process the data in batches.

## C. Datasets

*Table 6.* Statistics of multi-relational network datasets.

| Dataset | Nodes | Relation type | Edges | Features | Classes | Training | Validation | Test |
|---|---|---|---|---|---|---|---|---|
| ACM | 3,025 | Paper-Author-Paper (PAP)
Paper-Subject-Paper (PSP) | 26,416
2,197,556 | 1,902 | 3 | 600 | 300 | 2,125 |
| DBLP | 2,957 | Author-Paper-Author (APA)
Author-Paper-Conference-Paper-Author (APCPA) | 2,398
1,460,724 | 334 | 4 | 600 | 300 | 2,057 |
| Yelp | 2,614 | Business-User-Business (BUB)
Business-Service-Business (BSB)
Business-Rating Levels-Business (BLB) | 525,718
2,475,108
1,484,692 | 82 | 3 | 300 | 300 | 2,014 |
| MAG | 113,919 | Paper-Paper (PP)
Paper-Author-Paper (PAP) | 1,806,596
10,067,799 | 128 | 4 | 40,000 | 10,000 | 63,919 |
| Amazon | 11,944 | User-Product-User (UPU)
User-Star rating-User (USU)
User-Review-User (UVU) | 175,608
3,566,479
1,036,737 | 25 | 2 | 4,777 | 2,388 | 4,779 |

*Table 7.* Statistics of multimodal datasets.

| Dataset | Classes | Total | Features | Modal | Training | Validation | Test |
|---|---|---|---|---|---|---|---|
| IAPR | 6 | 7,855 | 100 | Image,text | 3,926 | 1,961 | 1,968 |
| ESP | 7 | 11,032 | 100 | Image,text | 5,514 | 2,754 | 2,764 |
| Flickr | 7 | 12,154 | 100 | Image,text | 6,076 | 3,037 | 3,041 |
| NUS | 8 | 20,000 | 100 | Image,text | 10,000 | 5,000 | 5,000 |

We choose 5 multi-relational network benchmark datasets in total. The statistics of the datasets are provided in Table 6. Following (Shen et al., 2024a), here we extract MAG from the original OGBN-MAG (Wang et al., 2020), remaining the nodes from the four largest classes.

We also select 4 multimodal datasets where the topological structure is not given. For a fair comparison, we adopt the identical data processing approach as described in (Mao et al., 2021) on all datasets, where the 4096-dimensional (D) image features are extracted by VGG-16 net and 768D text features are extracted by BERT net. To further improve efficiency, Principal Component Analyses (PCA) is employed to reduce the dimensions of both image features and text features to 100. The examples and descriptions of the aforementioned multimodal datasets are shown in Table 7.

## D. Proof

### D.1. Proof of theorem 5.1

*Proof.* Assume that $\mathcal{L}(\Theta g_i)$ can be divided into three parts: $\mathcal{L}_1$, $\mathcal{L}_2$, and $\mathcal{L}_3$, which are expressed as follows:

$$
\mathcal{L}_1 = \sum_{i=1}^{N} - \left( Y_i^\top log(\mathcal{S}(\Theta g_i)) \right), \mathcal{L}_2 = \sum_{i=1}^{N} \left( \gamma Y_i^\top \mathcal{S}(\Theta g_i) \right)
$$
$$
\mathcal{L}_3 = \gamma \sum_{i=1}^{N} \frac{1}{\alpha} log \left( \sum_{j=1}^{c} e^{\alpha(\mathcal{S}(\Theta g_i) - Y_i \odot \mathcal{S}(\Theta g_i))_j} \right)
$$

(16)

$\mathcal{L}_1$ and $\mathcal{L}_2$ are evidently convex with respect to $(\Theta g_i)$. Considering $\mathcal{L}_3$, the exponential term $\alpha(\mathcal{S}(\Theta g_i) - Y_i \odot \mathcal{S}(\Theta g_i))_j$ is a linear transformation with respect to $\mathcal{S}(\Theta g_i)$. Since the log-sum-exp function is well known to be convex, and the composition of a convex function with a linear function preserves convexity, the proof is complete.

## D.2. Proof of theorem 5.2

*Proof.* It is not difficult to prove that, through a variable substitution, the derivative $\frac{\partial \mathcal{L}_3}{\partial \Theta_{pq}}$ can be equivalently written as:

$$\frac{\partial \mathcal{L}_3}{\partial \Theta_{pq}} = \frac{\sum_{j=1}^{c} \mathcal{I}(j \neq m) e^{\alpha \mathcal{S}(\Theta g)_j} \mathcal{S}(\Theta g)_j (g_q \delta_{jp} - \mathcal{S}(\Theta g)_p g_q)}{\gamma^{-1}(\sum_{j=1}^{c} e^{\alpha \mathcal{S}(\Theta g)_j} - e^{\alpha \mathcal{S}(\Theta g)_m} + 1)} \tag{17}$$

Where $\mathcal{I}(j \neq m)$ is the indicator function, which is 0 when $j = m$ and 1 otherwise. It is evident that $|g_q| \leq 1$, $|\delta_{mp}| \leq 1$, and $|\mathcal{S}(\Theta g)_p g_q| \leq 1$. Therefore, the following inequality holds:

$$\left| \frac{\partial \mathcal{L}_1}{\partial \Theta_{pq}} \right| \leq |g_q \delta_{mp}| + |\mathcal{S}(\Theta g)_p g_q| \leq 2 \tag{18}$$

Similarly, we have:

$$\left| \frac{\partial \mathcal{L}_2}{\partial \Theta_{pq}} \right| \leq \gamma |\mathcal{S}(\Theta g)_m| \cdot |g_q \delta_{mp} - \mathcal{S}(\Theta g)_p g_q| \leq 2\gamma \tag{19}$$

Based on the fact that $|\sum_{j=1}^{c} e^{\alpha \mathcal{S}(\Theta g)_j} - e^{\alpha \mathcal{S}(\Theta g)_m} + 1| \leq c$, it is not difficult to prove that $\left| \frac{\partial \mathcal{L}_3}{\partial \Theta_{pq}} \right| \leq \frac{2\gamma e^{\alpha}}{c}$. For any $\Theta$ and $\tilde{\Theta}$, by the mean value theorem, there exists $\sigma$ such that:

$$\mathcal{L}(\Theta) - \mathcal{L}(\tilde{\Theta}) = \nabla \mathcal{L}(\sigma) \cdot (\Theta - \tilde{\Theta}) \tag{20}$$

Applying the Cauchy-Schwarz inequality, we obtain:

$$|\mathcal{L}(\Theta) - \mathcal{L}(\tilde{\Theta})| \leq \|\nabla \mathcal{L}(\sigma)\|_F \|\Theta - \tilde{\Theta}\|_F \tag{21}$$

where $\|\nabla \mathcal{L}(\sigma)\|_F^2 = \sum_{p,q} \left( \frac{\partial \mathcal{L}}{\partial \Theta_{pq}} \right)^2_{\Theta = \sigma}$. From the bounds on $\frac{\partial \mathcal{L}_3}{\partial \Theta_{pq}}$, we know:

$$\left( \frac{\partial \mathcal{L}}{\partial \Theta_{pq}} \right)_{\Theta = \sigma} \leq 2 \left( 1 + \gamma + \frac{\gamma e^{\alpha}}{c} \right) \tag{22}$$

Thus, summing over all $p, q$ gives:

$$\|\nabla \mathcal{L}(\sigma)\|_F^2 \leq \sum_{p,q} \left( 2 \left( 1 + \gamma + \frac{\gamma e^{\alpha}}{c} \right) \right)^2$$
$$= 4k^2 c \left( 1 + \gamma + \frac{\gamma e^{\alpha}}{c} \right)^2 \tag{23}$$

Taking the square root, the Lipschitz constant is therefore bounded by:

$$2k\sqrt{c} \left( 1 + \gamma + \frac{\gamma e^{\alpha}}{c} \right) \tag{24}$$

## D.3. Proof of theorem 5.3

*Proof.* Since the gradient of $\mathcal{L}(\Theta)$ is Lipschitz continuous with constant $L$, we have the inequality:

$$\mathcal{L}(\Theta_{t+1}) \leq \mathcal{L}(\Theta_t) + \langle \nabla \mathcal{L}(\Theta_t), \Theta_{t+1} - \Theta_t \rangle + \frac{L}{2} \|\Theta_{t+1} - \Theta_t\|^2 \tag{25}$$

Substituting the update rule $\Theta_{t+1} = \Theta_t - \eta \nabla \mathcal{L}(\Theta_t)$, we get:

$$\mathcal{L}(\Theta_{t+1}) \leq \mathcal{L}(\Theta_t) - \eta \|\nabla \mathcal{L}(\Theta_t)\|^2 + \frac{\eta^2 L}{2} \|\nabla \mathcal{L}(\Theta_t)\|^2 \tag{26}$$

Rearranging terms yields:

$$\mathcal{L}(\Theta_{t+1}) \leq \mathcal{L}(\Theta_t) - \left( \eta - \frac{\eta^2 L}{2} \right) \|\nabla \mathcal{L}(\Theta_t)\|^2 \tag{27}$$

To ensure a decrease in the objective function, choose $\eta \leq \frac{1}{L}$. For such $\eta$, the term $\eta - \frac{\eta^2 L}{2}$ is positive, and thus:

$$\mathcal{L}(\Theta_{t+1}) \leq \mathcal{L}(\Theta_t) - \frac{\eta}{2}\|\nabla\mathcal{L}(\Theta_t)\|^2 \tag{28}$$

Summing over $t = 0, 1, \ldots, T - 1$, we have:

$$\mathcal{L}(\Theta_0) - \mathcal{L}(\Theta_t) \geq \frac{\eta}{2}\sum_{t=0}^{T-1}\|\nabla\mathcal{L}(\Theta_t)\|^2 \tag{29}$$

Since $\mathcal{L}(\Theta)$ is lower-bounded by $\mathcal{L}^*$, it follows that:

$$\mathcal{L}(\Theta_0) - \mathcal{L}^* \geq \frac{\eta}{2}\sum_{t=0}^{T-1}\|\nabla\mathcal{L}(\Theta_t)\|^2 \tag{30}$$

Dividing through by $\eta T$ gives:

$$\frac{1}{T}\sum_{t=0}^{T-1}\|\nabla\mathcal{L}(\Theta_t)\|^2 \leq \frac{2(\mathcal{L}(\Theta_0) - \mathcal{L}^*)}{\eta T} \tag{31}$$

Thus, there exists some $t \in \{0, 1, \ldots, T - 1\}$ such that:

$$\|\nabla\mathcal{L}(\Theta_t)\|^2 \leq \frac{2(\mathcal{L}(\Theta_0) - \mathcal{L}^*)}{\eta T} \tag{32}$$

This proves the result.

## E. Methodology Details

Following the approach proposed in previous works (Liu et al., 2022; Shen et al., 2024b), after constructing the cosine similarity matrix of $H^v$, we implement post-processing techniques to ensure that $A'_v$ exhibits the characteristics of sparsity, non-negativity, symmetry, and normalization. For the convenience of discussion, the subscript $v$ is omitted hereinafter.

$K$**NN for sparsity.** In most applications, a fully connected adjacency matrix often has limited practical significance and incurs high computational costs. Therefore, we employ the $K$-Nearest Neighbors ($K$NN) operation to sparsify the learned graph. For each node, we retain the edges with the top-$K$ values and set the others to 0, thereby obtaining the sparse adjacency matrix $A^{sp}$. Note that we employ efficient $K$NN with locality-sensitive hashing (Fatemi et al., 2021) to enhance the model's scalability. This approach avoids the resource-intensive computation and storage of explicit similarity matrices, reducing the complexity from $\mathcal{O}(N^2)$ to $\mathcal{O}(NB)$, where $N$ is the number of nodes and $B$ is the batch size of the sparse $K$NN.

**Symmetrization and Activation.** Since real-world connections are typically bidirectional, we symmetrize the adjacency matrix. Additionally, the weight of each edge should be non-negative. Given the input $A^{sp}$, these operations can be expressed as follows:

$$A^{sym} = \frac{\sigma(A^{sp}) + \sigma(A^{sp})^\top}{2} \tag{33}$$

where $\sigma(\cdot)$ represents a non-linear activation implemented by the ReLU function.

**Normalization.** The normalized adjacency matrix with self-loops can be obtained as follows:

$$A' = (\tilde{D}^{sym})^{-\frac{1}{2}}\tilde{A}^{sym}(\tilde{D}^{sym})^{-\frac{1}{2}} \tag{34}$$

where $\tilde{D}^{sym}$ is the degree matrix of $\tilde{A}^{sym}$ with self-loops. Subsequently, for each view, we can acquire the adjacency matrix $A'_v$, which possesses the desirable properties of sparsity, non-negativity, symmetry, and normalization.

# F. Additional experiments

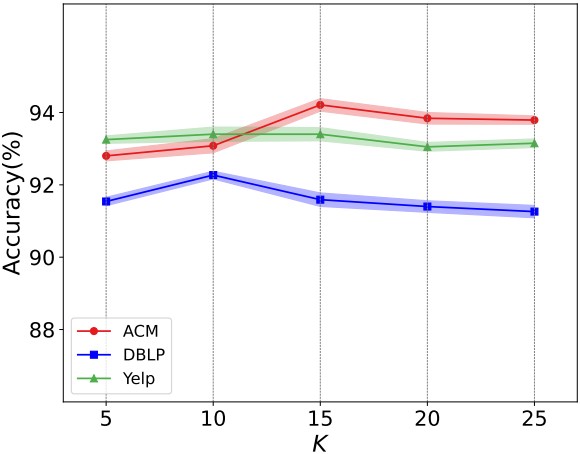

*Figure 5.* Sensitivity analysis on $K$.

Subsequent to the research on $\alpha$ and $\gamma$, an investigation into the sensitivity of the number of neighbors $K$ is conducted. The values of $K$ span a set of $\{5, 10, 15, 20, 25\}$. As depicted in Figure 5, despite the significance of $K$ as a model parameter, the CoE demonstrates minimal sensitivity to its variations. This finding suggests that the performance of the model remains relatively stable across a broad spectrum of $K$ values, thereby indicating its robustness in response to changes in this parameter.

