# OpenReview forum: "Cooperation of Experts: Fusing Heterogeneous Information with Large Margin"
_ICML.cc/2025/Conference — ICML 2025 poster_

### Official Review · Reviewer_nSKz · 2025-03-07

**Overall Recommendation:** 4

**Summary:**

The authors propose the Cooperation of Experts (CoE) framework to fuse heterogeneous information in multiplex networks. Authors design a two-level expert system where low-level experts focus on individual network layers while high-level experts capture cross-network relationships. The framework employs a large margin mechanism to optimize the collaboration among experts, ensuring that each expert's specialized knowledge is effectively utilized. The experiments demonstrate the superiority of their approach over state-of-the-art methods in both multi-relational and multi-modal tasks.

**Claims And Evidence:**

Clear and convincing evidence are given to support the claims.

**Essential References Not Discussed:**

None.

**Experimental Designs Or Analyses:**

I carefully reviewed the experimental setup and analyses to assess their rigor.

**Methods And Evaluation Criteria:**

The proposed method is well-suited for the heterogeneous information fusion problem. The datasets are representative of heterogeneous data, demonstrating the broad applicability of the method.

**Other Comments Or Suggestions:**

1."The cross-entropy loss Lcls(Z,Y ) is the upper bound of −I(G;Y ), where Z denotes the node representations of all nodes in network G."
Correction: "The cross-entropy loss Lcls(Z,Y) serves as an upper bound for −I(G;Y), where Z represents the node embeddings of all nodes in network G."

2. "We select two supervised structure-fixed GNNs—GCN (Kipf & Welling, 2016) and HAN (Wang et al., 2019)—as well as six supervised GSL methods."
Correction: "We select two supervised, structure-fixed GNNs—GCN (Kipf & Welling, 2016) and HAN (Wang et al., 2019)—along with six supervised GSL methods."

3."L(Θgi) is a convex function with respect to (Θgi)."
Correction: " L(Θgi) is a convex function with respect to Θgi."

**Other Strengths And Weaknesses:**

Pros：
1.	The motivation is clearly presented by Figure 1. The observed phenomenon is interesting and important.
2.	The expert perspective for multiplex network learning is interesting and novel.
3.	CoE is highly resilient to structural perturbations, which is crucial for real-world applications where data may be noisy or incomplete.

Cons：
The writing quality of the paper could be improved for better readability and clarity. Several grammatical errors, awkward phrasings, and unclear sentences make it difficult to follow the arguments.

**Questions For Authors:**

1.	The number of experts is given in Line 187. How do you get it? It should be explained in more detail.

2.	The supplementary material suggests that the authors add a loss convergence plot.

**Relation To Broader Scientific Literature:**

1. The paper introduces a novel framework that emphasizes expert cooperation rather than competition, which is a significant departure from traditional Mixture of Experts (MoE) approaches. This is a fresh perspective in the field of graph neural networks (GNNs) and multiplex network learning.

2. The large margin mechanism is an innovative optimization strategy that ensures robust expert collaboration, leading to improved predictive performance.

**Theoretical Claims:**

I reviewed the proofs and they appear to be correct.

---

> ### Author Rebuttal · Authors · 2025-03-31
>
> We sincerely thank the reviewer for the encouraging and constructive feedback. We greatly appreciate your recognition of our motivation, expert coordination design, and robustness to structural perturbations. Below we provide detailed responses to your suggestions. Figures and Tables are summarized in this link: **https://anonymous.4open.science/r/ICML_rebuttal-7D0E**.
>
> **Weakness & Suggestions: Writing and clarity**
>
> Thank you for pointing out the writing issues. We agree that improving readability is important for accessibility and impact. In the final version, we will:
>
> 1. Thoroughly proofread the entire manuscript for grammatical correctness and clarity.
>
> 2. Refine awkward phrasings, shorten overly long sentences, and ensure each technical statement is precisely and clearly expressed.
> 3. Review the supplementary materials and figure captions to maintain consistency and readability.
>
> We apologize for the writing quality issues in our manuscript, which may have caused some difficulty in following the arguments. Once again, we sincerely appreciate your careful reading, and we have made efforts to clarify the text in the revised version. We are confident that these improvements will significantly enhance the presentation quality of the paper.
>
> **Question1: Numbers of experts**
>
> Thanks for the concern, we appreciate the opportunity to clarify this point. In Section 4.2, we emphasize that the number of experts is fixed to $\frac{V(V+1)}{2}+1$, where $V$ is the number of views. This is mainly because CoE train $V$ low-level experts on each network view and $C^{2}_V= \frac{V(V-1)}{2}$ high-level experts.
>
> Additionally, we pointed out that an extra high-level expert is trained on $G_{tot}$. Thus, we have $V+ \frac{V(V-1)}{2}+1=\frac{V(V+1)}{2}+1$ experts in total. It is remarkable that the number of experts is limited, thus the overall expert training cost highly manageable. We analyze the model's efficiency **in the link above**.
>
> **Question 2: Loss convergence plot in the supplementary**
>
> Following your advice, we provide the convergence plots **in the link above**, on common-scale and large-scale datasets respectively.
>
> Across all datasets, the training loss decreases smoothly and steadily, without oscillation or divergence, reflecting stable optimization dynamics.  A significant loss drop is observed within the first 100–200 epochs, indicating that the confidence tensor and expert fusion strategy are effective at capturing informative gradients early in training. The convergence behavior is consistent across datasets of different scales and domains, which shows that our training process is not sensitive to specific data distributions or graph modalities. These properties collectively confirm that CoE is not only theoretically convergent (as shown in Theorem 5.5), but also empirically stable and efficient in practice.
>
> Once again, we sincerely appreciate your thoughtful comments and are encouraged by your support. Your feedback has helped us further improve the clarity and completeness of the paper. We hope the final version will fully meet your expectations, and we are more than happy to add clarifications to address any additional recommendations and reviews from you!

---

> > ### Comment · Reviewer_nSKz · 2025-04-04
> >
> > Thank you for addressing the key questions I raised. These supplementary clarifications have given me a more comprehensive understanding of the paper's value and significance. I will increase my rating for this paper.

---

### Official Review · Reviewer_t4sH · 2025-03-09

**Overall Recommendation:** 3

**Summary:**

This paper proposes the Cooperation of Experts (CoE) framework, which solves the problem of multimodal heterogeneous information fusion by constructing a heterogeneous multiplexing network. The research focuses on the challenge of pattern heterogeneity across semantic spaces, designing specialized encoders as domain experts, combining large interval collaboration mechanisms and optimization strategies to achieve robust modeling and knowledge complementary extraction of complex data structures.

**Claims And Evidence:**

Yes.

**Essential References Not Discussed:**

> Common ensemble methods include bagging (Zhou & Tan, 2024) and boosting (He et al., 2024).

In recent years, deep forests [Zhou and Feng, NSR 2019] have emerged in the field of ensemble learning, combining bagging and boosting techniques. Especially the theoretical analysis of its large margin property [Lyu et al., NeurIPS 2019] is highly relevant to this paper.

1. Zhou, Z. H., & Feng, J. (2019). Deep forest. National Science Review, 6(1), 74-86.
2. Lyu, S. H., Yang, L., & Zhou, Z. H. (2019). A refined margin distribution analysis for forest representation learning. Advances in Neural Information Processing Systems, 32.

**Experimental Designs Or Analyses:**

Yes.

**Methods And Evaluation Criteria:**

Yes.

**Other Comments Or Suggestions:**

NAN.

**Other Strengths And Weaknesses:**

Strengths:
* For the first time, a framework emphasizing expert cooperation rather than competition was proposed, breaking through the paradigm limitations of expert competition in traditional MOE models. Introducing the large margin optimization mechanism into expert collaboration scenarios provides a new perspective for model optimization (the theoretical innovation of this mechanism is validated by comparing it with traditional RF/WRF methods).
* The experimental design covers 0-90% of the network structure disturbance intensity and systematically verifies the robustness of the method in extreme attack scenarios, making up for the shortcomings of existing research on high-intensity disturbance testing.

Weaknesses:
* Theoretical depth limitation: No mathematical convergence proof or generalization error boundary analysis provided for the large margin mechanism.
* Doubtful generalization of scenarios: The current experiment is only based on the ACM dataset and has not validated its effectiveness on larger scale graph data or cross modal scenarios.

**Questions For Authors:**

NAN.

**Relation To Broader Scientific Literature:**

* Inspired by the MoE (Mixture of Experts) framework, but innovatively introducing a learnable confidence tensor, it solves the limitation of expert competition rather than cooperation in traditional MoE (such as the unactivated expert utilization problem proposed by Shi et al., 2024a).
* By maximizing the margin of the predicted results, the theory of ensemble learning (such as Boosting's weight adjustment) is combined with graph neural networks, distinguishing it from static expert weight allocation methods (such as Liu et al., 2022's fixed gating mechanism).

**Theoretical Claims:**

Yes.

---

> ### Author Rebuttal · Authors · 2025-03-31
>
> We appreciate your thoughtful feedback. Your constructive criticism is invaluable in refining our work. Below, we give point-by-point responses to your comments.
>
> **Weakness 1**
>
> We fully agree that deeper theoretical analysis strengthens the credibility of a new learning framework. To clarify, we would like to emphasize that **Theorem 5.5 in the main paper provides a mathematical convergence analysis** of the optimization procedure. This theorem rigorously proves that the gradient norm of training objective is guaranteed to converge to zero at a sublinear rate, which establishes the optimization stability and convergence of our training process. Following your advice, we further prove a **generalization error bound** ---- a probabilistic upper bound on the 0-1 loss of the CoE classifier:
>
> Let $\mathcal{X}\times\mathcal{Y}$ be the input-label space with $|\mathcal{Y}|=C$ classes. We have an i.i.d. training sample $S=\{(x_i,y_i)\}_{i=1}^n$.
>
> A CoE classifier $f$ produces a probability vector over $C$ classes, denoted $f(x)=(f_1(x),\dots,f_C(x))^\top$. We define the margin of $f$ at $(x,y)$ as
> $$
> \gamma_f(x,y):=f_y(x)- \max_{y'\neq y} f_{y'}(x),
> $$
> a large positive $\gamma_f(x,y)$ implies a strong preference for the correct class $y$. The usual 0-1 loss is $\ell_{\mathrm{0\text{-}1}}(f;x,y):=\mathbb{I} [\arg\max_{c}f_c(x)\neq y].$ We also define $\ell_{\gamma}^{\mathrm{0\text{-}1}}(f;x,y):=\mathbb{I} [\gamma_f(x,y)\leq0]$ and the ramp loss:
>
> $$
> \ell_{\gamma}(f;x,y) :=
> \begin{cases}
>   0, & \gamma_f(x,y)\ge \gamma,\\\\
>   1-\dfrac{\gamma_f(x,y)}{\gamma}, & 0<\gamma_f(x,y)<\gamma,\\\\
>   1, &\gamma_f(x,y)\le0.
> \end{cases}
> $$
> One has $\ell_{\mathrm{0\text{-}1}}(f;x,y)\le \ell_{\gamma}^{\mathrm{0\text{-}1}}(f;x,y) \le \ell_{\gamma}(f;x,y).$
>
> Let $\ell_\gamma\circ\mathcal{F}$ be the set of ramp-loss functions induced by a hypothesis class $\mathcal{F}$, where each $f\in\mathcal{F}$ is a CoE classifier. Then for any $\delta>0$, with probability at least $1-\delta$ over an i.i.d. sample $S=\{(x_i,y_i)\}_{i=1}^n$, the following holds for all $f\in\mathcal{F}$:
>
> $$
> \mathbb{E}[\ell _{0-1}(f)]\le \mathbb{E} _{(x,y)\sim\mathcal{D}}[\ell _\gamma^{0-1}(f;x,y)]\le \frac{1}{n}\sum _{i=1}^n\ell _\gamma(f;x _i,y _i)+\frac{2}{\gamma}\mathfrak{R} _n(\mathcal{F})+3\sqrt{\frac{log(\frac{2}{\delta})}{2n}},
> $$
>
> where $\mathfrak{R}_n(\mathcal{F})$ is the Rademacher complexity of the CoE margin-function class. Due to space limitations, the detailed proof will be included in the final version and is omitted here.
>
> In CoE framework, we have $k$ experts $E_1,\dots,E_k$, each outputs a probability vector $E_j(x)\in \mathbb{R}^C$. Besides, we have a confidence tensor $\Theta$ and we form $g(x)=[E_1(x)^\top,\dots,E_k(x)^\top]^\top$, then $f(x)=\mathrm{softmax}\bigl(\Theta g(x)\bigr).$ And the margin is $\gamma _f(x,y)=[\Theta g(x)] _y-\max _{y'\neq y}[\Theta g(x)] _{y'}$.
>
> We assume $||\Theta|| _F\le B _\Theta$ and $||E _j(x)|| _2\le G _e ,\forall j,x$. Hence $||g(x)|| _2\le \sqrt{k}G _e$. Let $\mathcal{F} _\Theta$ be the set of CoE margin functions $\gamma_f$.
>
> Then $\mathfrak{R} _n(\mathcal{F} _\Theta)\le C _\mathrm{MC} \frac{B _\Theta G _e \sqrt{k}}{\sqrt{n}}$, where $C _\mathrm{MC}$ is a constant on the order of $\sqrt{ln(C)}$, reflecting the multi-class max operation.
>
> With probability at least $1-\delta$, all $f \in \mathcal{F}_ \Theta$ satisfy
>
> $$
> \mathbb{E}\bigl[\ell_{\mathrm{0\text{-}1}}(f)\bigr]\le\frac1n \sum_{i=1}^n \ell_{\gamma}\bigl(f;x_i,y_i\bigr)+\frac{2B_\Theta G_e\sqrt{k}}{\gamma \sqrt{n}}+3\sqrt{\frac{\log(\tfrac2\delta)}{2n}}.
> $$
>
> It shows that ensuring a large margin $\gamma$ and controlling the norms $B_\Theta$ (confidence-tensor magnitude) and $G_e$ (expert-output scale) leads to a generalization guarantee. Increasing the number of experts $k$ has a $\sqrt{k}$ impact, illustrating the trade-off between model capacity and margin-based guarantees. Due to CoE mechanism limits the number of $k$ and $B_\Theta$,  while $G_e$ is fixed, thus our model has remarkable generalization ability.
>
> **Weakness 2**
>
> We would like to clarify that our main experiments already include evaluations on **large-scale network datasets** such as Amazon and MAG in Table 1, all of which involve diverse views and heterogeneous information types. Besides, we conduct experiments on four **multi-modal datasets** in Table 2. To further strengthen our claims, we conduct robustness experiments on large-scale datasets and main experiment on an additional large-scale dataset DGraph, results are concluded in **https://anonymous.4open.science/r/ICML_rebuttal-7D0E**.
>
> Once again, we appreciate your valuable suggestions to help us improve our work. We will include the missing citations in the revised version and ensure all relevant work be properly cited. With the newly added theoretical analysis and experiments, we hope your concerns have been addressed. We sincerely hope this response and the revised improvements can earn your stronger recommendation.

---

### Official Review · Reviewer_s8EY · 2025-03-10

**Overall Recommendation:** 4

**Summary:**

This paper proposes the Cooperation of Experts (CoE) framework, which aims to address the challenge of fusing heterogeneous information in modern data analysis. The CoE framework encodes multi-typed information into unified heterogeneous multiplex networks and allows dedicated encoders, or "experts," to collaborate rather than compete. The authors claim that this approach captures the intricate structures of real-world complex data and outperforms existing methods.

**Claims And Evidence:**

Theoretical analysis and extensive experimental on multi-relational graphs and multi-modal data verify the claims.

**Essential References Not Discussed:**

The references are complete.

**Experimental Designs Or Analyses:**

The experiments are conducted on two types of data and compared with different categories of methods, which is sufficient.

**Methods And Evaluation Criteria:**

The proposed method makes sense to me. The experiments are conducted on two types of data, verifying the generalizability of the proposed method.

**Other Comments Or Suggestions:**

1. The limitation of the proposed method should be discussed.
2. "While boosting relies on sequential training where each model builds upon the previous one"->"While boosting relies on sequential training, where each model builds upon the previous one"
3. Some notations are confusion. For example, L represents different meanings in the supplementary.

**Other Strengths And Weaknesses:**

Strengths:
1. The CoE framework introduces a unique approach to handling heterogeneous information by transcending modality and connection differences, which is a significant advancement in the field.
2. The paper provides rigorous theoretical analyses to support the feasibility and stability of the CoE framework, adding credibility to the proposed method.
3. The extensive experiments across diverse benchmarks demonstrate the superior performance of the CoE framework, indicating its broad applicability and effectiveness.

Weaknesses
1. While the paper verifies the effectiveness of the CoE framework, it lacks a deeper discussion on the scalability of the approach, especially for very large datasets. Training multiple experts and computing mutual information across networks introduces computational complexity.
2. The paper could benefit from some experimental analysis in terms of computational efficiency, as this is an important aspect of practical applications.

**Questions For Authors:**

1. How does the CoE framework perform in terms of scalability when applied to very large datasets? Are there any specific challenges or limitations?
2. Can the authors provide more details on the computational efficiency of the CoE framework in the experiments?

**Relation To Broader Scientific Literature:**

The proposed CoE framework encodes multi-typed information into unified heterogeneous multiplex networks, which transcends modality and connection differences. It provides a unified approach to handle real-world complex data. To my best knowledge, this is the first application of expert learning idea to multiplex networks.

**Theoretical Claims:**

Theoretical justification is provided to show the convergence property of the proposed method, which makes sense to me.

---

> ### Author Rebuttal · Authors · 2025-03-31
>
> We appreciate your thoughtful feedback. Your constructive criticism is invaluable in refining our work. Below, we give point-by-point responses to your comments. Figures and Tables are summarized in this link: **https://anonymous.4open.science/r/ICML_rebuttal-7D0E**.
>
> **Weakness 1 & Question 1: Model scalability**
>
> We appreciate your important concern about scalability. CoE is designed to be modular and extensible, and we agree that scalability becomes critical for real-world deployment. In fact, the following characteristics of CoE are particularly favorable for large-scale data:
>
> Parallelizable Expert Modules: Unlike boosting or sequential learning schemes, experts are trained in parallel, which makes the framework highly suitable for distributed training and parallel inference.
>
> Expert Fusion via Lightweight Confidence Tensor: Rather than using complex hierarchical gating or stacking mechanisms, CoE fuses the predictions of all experts through a simple yet effective linear tensor-based confidence fusion mechanism. This module has negligible memory and computational overhead, and scales linearly with the number of experts.
>
> Reformulating Mutual Information (MI): network-level MI is replaced by node-level representations to reduce complexity.
>
> Scalability-Aware Experiments: On the largest datasets in our experiments (e.g., MAG (113,919 nodes) and Amazon (11,944 nodes)), CoE demonstrates not only strong performance but also comparable training time to state-of-the-art baselines (see the next part).
>
> New Large-Scale Benchmark: We additionally include a new large-scale experiment on the DGraph dataset, with 111,310 nodes and 430k+ edges. CoE outperforms other scalable baselines such as InfoMGF and other baselines which get relatively high scores on the previous datasets. These results provide strong empirical evidence that CoE can scale to realistic large-scale graph settings. To directly show the performance, we summarize the results on MAG, Amazon and DGraph **in the link above**.
>
> **Weakness 2 & Question 2: Empirical computational complexity**
>
> We appreciate the reviewer’s concern regarding the computational efficiency of CoE. Although CoE adopts a two-stage structure where experts are trained individually before being fused, we emphasize that:
>
> - **The number of experts is limited** and does not grow with data size. In practice, we only train limited number of experts, which makes the overall expert training cost highly manageable.
> - **Fast convergence is achieved**. This is achieved due to the design of our confidence tensor and the optimization strategy. Specifically, the confidence tensor is a lightweight linear transformation trained jointly with the final fusion stage, and it converges rapidly in practice.
> - **Linear scalability is observed** on large-scale datasets. Despite the two-stage structure, CoE exhibits empirically near-linear runtime scaling with dataset size. For example, on large-scale datasets like MAG and DGraph, CoE trains even faster than several single-encoder GSL baselines, as detailed **in the link above**. This demonstrates that the proposed architecture does not incur significant overhead compared to standard GNN-based models.
>
> **Suggestion 1: Limitation of proposed method**
>
> Currently, CoE is designed for heterogeneous multiplex networks, the current formulation assumes static input structures. Extending the framework to accommodate more complex graph settings like dynamic or hierarchical networks remains an interesting direction for future research. Additionally, although we adopt GCN as the base encoder in this work for fairness in comparison, incorporating other architectures such as GAT or graph transformers remains future work.
>
> **Suggestion 2: "While boosting relies on sequential training where each model builds upon the previous one"->"While boosting relies on sequential training, where each model builds upon the previous one"**
>
> We sincerely appreciate the suggestion. We will revise the corresponding phrase in the refined version.
>
> **Suggestion 3: Notation confusion**
>
> Sorry for the trouble caused by our symbols, we apologize for the confusion caused by inconsistent notation. As you correctly noted, some symbols such as $\mathcal{L}$ are overloaded in the supplementary, mainly occurred in Section D.  We will unify and revise all instances of $\mathcal{L}$ in the final version of the paper. Additionally, we will conduct a thorough pass of the paper to ensure precise and consistent mathematical expression.
>
> Your concerns about scalability and efficiency are well-justified and have helped us greatly improve the clarity and practicality of our method. We hope the new large-scale experiments, added analysis, and revisions in the final version will adequately address your concerns. We sincerely hope this response convinces you that CoE is scalable, practical, and broadly applicable, and that our efforts in addressing your suggestions merit a higher overall recommendation.

---

> > ### Comment · Reviewer_s8EY · 2025-04-06
> >
> > Thanks to the authors for the detailed response to the concerns. I have no other concerns and think this work brings a clear contribution to the related community. So I keep the positive rating.

---

### Official Review · Reviewer_LQq1 · 2025-03-12

**Overall Recommendation:** 4

**Summary:**

This paper presents the CoE framework, a groundbreaking method for extracting knowledge from diverse and multi-layered networks. Its core novelty lies in a hierarchical expert coordination mechanism, where specialized low-level experts focus on capturing unique relational patterns, while high-level experts integrate insights from across these networks. The framework is further refined through the incorporation of a large margin mechanism, which optimizes expert collaboration, enhancing both robustness and generalization capabilities. Theoretical assessments validate the feasibility of the proposed method, and comprehensive experiments conducted on benchmark datasets demonstrate its superiority compared to existing techniques.

**Claims And Evidence:**

Yes, they possess a convincing persuasive force.

**Essential References Not Discussed:**

No.

**Experimental Designs Or Analyses:**

Yes, I have checked the validity of the experimental designs and analyses.

**Methods And Evaluation Criteria:**

Yes, the methodologies and assessment standards put forth should be pertinent and suitable for the problem or application in question.

**Other Comments Or Suggestions:**

1) A visualization or an explainability analysis (e.g., SHAP values) would be useful.
2) Remove the "..., etc." as "etc." already implies continuation in Line 48.
3) “(possible with different attributes) but a different type of links” should be changed to “(possibly with different attributes) but different types of links”.
4) "The symbol “+" denotes directly add up the networks" can be changed to "The symbol “+” denotes directly adding the networks".

**Other Strengths And Weaknesses:**

Strengths:
1) Novel Expert Coordination Strategy: (a) Unlike prior Mixture of Experts (MoE) models that rely on gating mechanisms (activating a subset of experts), CoE promotes cooperation instead of competition. (b) The introduction of high-level experts allows for cross-network knowledge fusion, enhancing model flexibility.
2) Strong Theoretical Foundation: (a) The paper provides convexity and Lipschitz continuity proofs, ensuring convergence and stability. (b) The mutual information maximization strategy enhances the fused representation’s effectiveness. (c) The confidence tensor mechanism allows all experts to contribute dynamically, preventing over-reliance on a small subset.
3) State-of-the-Art Performance: CoE outperforms all baselines on multiplex network and multimodal classification tasks.

Weaknesses:
1) Interpretability of Expert Decisions: While CoE optimizes expert collaboration, it does not provide an explicit mechanism to interpret the contributions of individual experts. A visualization or an explainability analysis (e.g., SHAP values) would be useful.

**Questions For Authors:**

1) The paper primarily compares CoE with graph structure learning (GSL) and MoE models. Are there any attention-based approaches (e.g., Transformer-based graph fusion)?
2) How does CoE handle conflicting expert opinions? Does the large-margin mechanism still work?

**Relation To Broader Scientific Literature:**

In this work, unlike prior Mixture of Experts (MoE) models that rely on gating mechanisms (activating a subset of experts), CoE promotes cooperation instead of competition. The application of MoE to heterogeneous information fusing is new.

**Theoretical Claims:**

Yes, I have checked the validity of the proofs underpinning the theoretical claims discussed.

---

> ### Author Rebuttal · Authors · 2025-03-31
>
> We appreciate your thoughtful feedback. We are especially grateful for the positive recognition of our novel expert coordination strategy, solid theoretical foundation, and strong empirical performance. Below, we provide responses to the specific suggestions and questions. Figures and Tables are summarized in this link: **https://anonymous.4open.science/r/ICML_rebuttal-7D0E**
>
> **Weakness & Suggestion1: Interpretability of Expert Decisions**
>
> We appreciate your suggestion regarding the interpretability of expert contributions. In fact, we have emphasized the relative contribution of each expert using confidence scores in Figure 1(a) and 1(b) of the main paper, which demonstrate how different experts participate dynamically across semantic contexts.
>
> In addition, to enhance interpretability more explicitly, we further compute SHAP values for each expert across different datasets. For better visualization, we normalize the SHAP values within each dataset and display them in tables for small-scale expert settings. For datasets with a larger number of experts, we provide heatmaps to show expert influence across different classes, enabling a more fine-grained interpretation of their roles.
>
> These analyses confirm that our confidence tensor indeed reflects meaningful and diverse expert specializations. We include these SHAP-based visualizations and interpretation discussions **in the link above**.
>
> **Suggestion2 & 3 & 4: Corrections of writings**
>
> Sorry for the mistake we made. We thank the reviewer for pointing out the writing issues and ambiguous expressions. We will carefully revise the identified sentences to improve clarity and precision. Specifically, we will rephrase the description of the “+” and “etc.” symbols for better readability, and correct the noted grammatical issues. These improvements will be incorporated into the revised version to enhance overall presentation quality.
>
> **Q1: Attention-based approaches**
>
> We apologize for not clearly highlighting the attention-based baselines in the original submission. Thank you for raising this point. In fact, among our baselines, NodeFormer (Wu et al., 2022) and HAN (Wang et al., 2019) are attention-based methods. To strengthen our comparison and address this oversight, we additionally include SGFormer [1], a recent Transformer-based graph structure learning method that applies self-attention over both features and learned graph topology. We compare its performance with CoE on all datasets, and present the results **in the link above**. This updated comparison confirms that CoE consistently outperforms attention-based methods across all datasets.
>
> [1] Wu et al. "SGFormer: Simplifying and Empowering Transformers for Large-Graph Representations." NeurIPS (2023).
>
> **Q2: How does CoE handle conflicting expert opinions?**
>
> The confidence matrix $\Theta$ we learn represents the authority of each expert. To explain with a simple example: if expert $i$ believes the sample belongs to class $p$, the distribution vector of their judgment is $\Theta(:,p,i)$. If another expert $j$ believes it belongs to class $q$, the distribution vector is $\Theta(:,q,j)$. The combined opinion of the two experts is $ \Theta(:,p,i) + \Theta(:,q,j)$. The final classification result is then obtained through Eq. (5): $\hat{y}_i=\underset{j = 1...c}{argmax}\ \left(\mathcal{S}\left(\Theta g_i\right)\right)_j$.
>
> Each expert provides an opinion, which we consider as a probability distribution vector. By summing the probability distribution vectors, normalizing the result, and performing an argmax operation, we can handle both similar and opposing opinions among the experts.
>
> Once again, we sincerely appreciate your time and effort in reviewing our paper. Your constructive criticism has been invaluable in refining our work, and we are more than happy to add clarifications to address any additional recommendations and reviews from you!

---

### Decision · Program_Chairs · 2025-05-01

**Decision:**

Accept (poster)

**Comment:**

This paper proposes a groundbreaking method for extracting knowledge from diverse and multi-layered networks. All reviewers give positive scores. The motivation is clear. The writing is smooth. The experiment is sufficient.